# A Computational Framework for Prediction and Analysis of Cancer Signaling Dynamics from RNA Sequencing Data—Application to the ErbB Receptor Signaling Pathway

**DOI:** 10.3390/cancers12102878

**Published:** 2020-10-07

**Authors:** Hiroaki Imoto, Suxiang Zhang, Mariko Okada

**Affiliations:** 1Institute for Protein Research, Osaka University, Suita, Osaka 565-0871; Japan; himoto@protein.osaka-u.ac.jp (H.I.); motoka-zhang@protein.osaka-u.ac.jp (S.Z.); 2Center for Drug Design and Research, National Institutes of Biomedical Innovation, Health and Nutrition, Ibaraki; Osaka 567-0085, Japan; 3Institute for Chemical Research, Kyoto University, Kyoto 611-0011, Japan

**Keywords:** mathematical modeling, parameter estimation, ErbB signaling pathway, breast cancer

## Abstract

**Simple Summary:**

Temporal signaling dynamics are important for controlling the fate decisions of mammalian cells. In this study, we developed BioMASS, a computational platform for prediction and analysis of signaling dynamics using RNA-sequencing gene expression data. We first constructed a detailed mechanistic model of early transcriptional regulation mediated by ErbB receptor signaling pathway. After training the model parameters against phosphoprotein time-course datasets obtained from breast cancer cell lines, the model successfully predicted signaling activities of another untrained cell line. The result indicates that the parameters of molecular interactions in these different cell types are not particularly unique to the cell type, and the expression levels of the components of the signaling network are sufficient to explain the complex dynamics of the networks. Our method can be further expanded to predict signaling activity from clinical gene expression data for in silico drug screening for personalized medicine.

**Abstract:**

A current challenge in systems biology is to predict dynamic properties of cell behaviors from public information such as gene expression data. The temporal dynamics of signaling molecules is critical for mammalian cell commitment. We hypothesized that gene expression levels are tightly linked with and quantitatively control the dynamics of signaling networks regardless of the cell type. Based on this idea, we developed a computational method to predict the signaling dynamics from RNA sequencing (RNA-seq) gene expression data. We first constructed an ordinary differential equation model of ErbB receptor → c-Fos induction using a newly developed modeling platform BioMASS. The model was trained with kinetic parameters against multiple breast cancer cell lines using autologous RNA-seq data obtained from the Cancer Cell Line Encyclopedia (CCLE) as the initial values of the model components. After parameter optimization, the model proceeded to prediction in another untrained breast cancer cell line. As a result, the model learned the parameters from other cells and was able to accurately predict the dynamics of the untrained cells using only the gene expression data. Our study suggests that gene expression levels of components within the ErbB network, rather than rate constants, can explain the cell-specific signaling dynamics, therefore playing an important role in regulating cell fate.

## 1. Introduction

Cancer is considered a genetic and lifestyle disease. Signaling-related genes are often altered in a genetic and epigenetic manner, and those changes are encoded in the spatiotemporal dynamics of a signal network [1]. To infer underlying mechanisms involved in the dysregulation of cancer signaling networks, mathematical modeling has emerged as a powerful method [2,3,4,5]. Specifically, the ordinary differential equation (ODE) model is a valuable tool that allows us to understand biological systems in a quantitative manner and to generate insights into underlying regulatory mechanisms to allow for potential therapeutic manipulation of the system [6]. ErbB receptor signaling is one of the prototypical signaling pathways involved in a variety of physiological processes and its dysregulation is tightly associated with the development and progression of many types of human cancers [7]. Systems-level analysis of ErbB receptor signaling pathways has revealed ligand-specific cellular responses arising from this network and the pathway control mechanisms [8,9,10]. Nonetheless, it remains poorly understood why the same stimulus (such as a growth factor) evokes completely different responses in different cell types. To clarify the mechanism, we used a mechanistic model to reveal the relationship between genetic information in different cell lines and their ErbB signaling dynamics.

The ErbB receptor tyrosine kinase family is composed of four members, ErbB1 (also called epidermal growth factor (EGF) receptor), ErbB2, ErbB3, and ErbB4, and there are several ligands known to bind to these receptors. Upon ligand binding, the receptors form dimers, activate their intrinsic tyrosine kinases and stimulate multiple signaling pathways [11]. With this sequence of events, cells transmit information from the extracellular environment to the nucleus via the temporal activation patterns of signaling molecules [12]. For example, EGF and another member of the EGF family, heregulin (HRG), induce transient and sustained ERK activity in cell lines such as MCF-7 that express low and high levels of ErbB1 and ErbB3 receptors, respectively [13,14]. Ultimately, these differences in ERK activation dynamics change expression patterns of immediate-early genes, such as c-Fos and c-Myc transcription factors [15], which play a central role in controlling cell cycle [16,17], differentiation [13,18], and metabolism [19]. Therefore, a system-level understanding of the common mechanisms that regulate ErbB dynamics across cell types is important for uncovering a fundamental mechanism in individual cancer progression.

In this study, we constructed a comprehensive mathematical model of the ErbB receptor to c-Fos activation signaling cascade by integrating two previously developed independent mathematical models describing the processes of ErbB receptor signaling [20] and c-Fos transcriptional regulation [14]. However, during this process, we found that the integration of independently developed mathematical models is not simple. Because an integrated model becomes extremely large, the network contains multiple non-linear regulatory mechanisms such as feedback loops [21,22,23,24,25], and the available experimental data is limited. Therefore, it was difficult to obtain reproducible parameters (e.g., rate constants of enzymes and protein interactions) by minimizing an objective function, i.e., the distance between simulated values and experimental data [26]. To deal with this problem, we developed a new modeling platform, BioMASS (Modeling and Analysis of Signaling Systems), tailored to parameter estimation and sensitivity analysis of complex biological systems. The BioMASS framework allows efficient optimization of multiple parameter sets simultaneously and generates the multiple parameter candidates that explain the signaling dynamics of interest. These parameter candidates can be further evaluated by their distribution and sensitivity analysis as a part of alternative information about the hidden regulatory mechanism of the system. This type of analysis tool is particularly useful to elucidate system behaviors when the network of interest is relatively large. 

We utilized BioMASS for parameter training on cancer cell lines and hypothesis generation concerning the regulatory principles of the ErbB signaling network. We first trained the kinetic parameters of the ErbB network against phosphorylation data from multiple breast cancer cell lines together with their autologous RNA-seq data obtained from the Cancer Cell Line Encyclopedia (CCLE) to infer the initial protein level of the model species. Then, the model proceeded to prediction in another cell line on which it was not trained using its own phosphorylation data. As a result, the model learned the parameters from other cells lines and was able to accurately predict the quantitative behavior of the untrained cell line solely from its autologous gene expression data. This result indicates a possibility that cell-type specific ErbB signaling dynamics can be explained by gene expression levels, not by differences in the kinetic parameters in the network. Accordingly, it should be possible to predict the dynamic properties of a signaling network only from public transcriptome data, and our computational framework will be useful to perform the analysis for identification of the key regulators and for assessment of therapeutics targeting the network.

## 2. Results

### 2.1. Development of BioMASS, a Framework for Modeling, Simulation, and Parameter Estimation

Mechanistic modeling is a powerful method to investigate complex biological systems [27,28] since it enables one to integrate prior knowledge in the research field with different datasets generated under a wide range of experimental conditions. Although a significant number of modeling studies are dedicated to explaining the dynamic properties of cellular signaling, most of the biological processes described in these models are limited and do not comprehensively encompass signaling pathways in mammalian cells. Therefore, a current challenge is to integrate these independently constructed models and thus to understand a signaling system at the whole cell level. However, this task can succeed only by overcoming some problems, e.g., as the model becomes larger, parameter estimation of the model to reproduce experimental observations, such as dose response and time series data, becomes more difficult. 

To overcome this problem, we developed BioMASS (https://github.com/okadalabipr/biomass), a Python framework for Modeling and Analysis of Signaling Systems (Figure 1a). BioMASS is a user-friendly simulation tool for experimental biologists and currently implements the model of Nakakuki et al. [14] as a showcase (Figure 1b,e). BioMASS supports (1) parameter estimation of ODE models (Figure 1c), (2) sensitivity analysis (Figure 1d) and (3) effective visualization of simulation results (Figure 1f,g).

In the modeling of biological systems, parameter estimation is one of the most critical steps [29]. Nevertheless, this approach in biological applications has several weak points. Since differential equations describing biological networks are often complex and the solution space is non-convex and non-linear, the ODE optimization problem can contain multiple local optima. In addition, experimental datasets used for the parameter estimation are often sparse and not continuous. To solve this problem, BioMASS enables integrative evaluation of parameter estimation and sensitivity analysis of the dozens of parameter candidates obtained by data fitting. We implemented a genetic algorithm [30,31,32,33], one of the efficient methods for solving global optimization problems for the data fitting of the model. After model fitting, users can perform sensitivity analysis to identify critical parameters, species or regulations in the system of interest. By assessing parameter fitting and sensitivity analysis together, it is possible to identify critical or robust regions in the network, or the origin of heterogeneity generated from the network.

### 2.2. Development of Comprehensive Model of the Immediate-Early Gene Response Triggered by the ErbB Receptor in Four Breast Cancer Cell Lines

In a previous study, a mathematical model accounting for c-Fos regulation in response to ErbB receptor activation was developed to understand how different types of growth factors give rise to unique gene expression patterns and cell fates [14]. The model provided novel insight into the control mechanism by which cells convert transient and sustained ERK signals into an all-or-none response of phosphorylated c-Fos. However, the input for the previous c-Fos model was derived from an interpolation of the phosphorylated MEK signal quantified from Western blots of the MCF-7 cell line stimulated with two types of growth factors.

In this study, to investigate the common regulatory mechanisms of ErbB receptor signaling on transcriptional control in multiple breast cancer cell lines, we integrated and constructed a comprehensive model of ErbB signal transduction [20] and the immediate-early gene response [14] (Figure 2). The integrated model contains several important processes in cancer, such as the MAPK pathway, the PI3K-Akt pathway and c-Fos induction for cell proliferation and survival. We considered phosphorylated ERK (pERK), Akt (pAkt), and c-Fos (pc-Fos) to be key determinants of this network and measured time course patterns of these proteins in four breast cancer cell lines, MCF-7, BT-474, SK-BR-3, and MDA-MB-231, which represent different breast cancer subtypes, Luminal A, Luminal B, HER2-positive and Triple-negative, respectively [34].

### 2.3. Training Model Parameters Using Gene Expression Data

For construction of a cell-specific model for each cell line, we incorporated gene expression data from the Cancer Cell Line Encyclopedia (CCLE) [35] to inter the initial values of the 19 gene products (Figure 2 and Appendix A for full gene list) described in the model. To make a model individualized to particular cell lines, Fröhlich et al. [36] assumed that kinetic parameters, such as transport, binding, and phosphorylation rates, depend only on the chemical properties of the involved species and are identical across cell lines. In addition, genome-wide association studies (GWAS) have shown that disease-related genomic mutations are significantly associated with gene expression levels in *cis* or *trans*-acting genes and have been named Expression Quantitative Trait Loci (eQTL) [37]. We assume that differences in rate constants, which may originate from mutations in protein coding regions, will be negligible when compared to the gene expression levels of the cancer network. Therefore, this study introduced the same assumption of Fröhlich et al. to infer the ErbB signaling dynamics. 

We used time series data of pAkt, pERK, and pc-Fos from three breast cancer cell lines (MCF-7, BT-474 and MDA-MB-231) and autologous CCLE RNA-seq expression data to train the model parameters (Appendix A). To extract the gene expression datasets of interest from CCLE and input them to the BioMASS platform, we also developed a tool, ccle_extractor (https://github.com/okadalabipr/ccle_extractor). To estimate parameters using the training datasets, we minimized the residual sum of squares to obtain common parameter sets that can reproduce ErbB signaling dynamics in these three cell lines. We ran optimization simultaneously and finally obtained 30 good-fitting parameters (Figure 3, Appendix A).

### 2.4. Model-Based Prediction of ErbB Signaling Dynamics

To test our hypothesis that the gene expression level is the major factor that determines the network dynamics, we attempted to predict signaling dynamics in another breast cancer cell line subtype, SK-BR-3, from its gene expression data. Since we assumed that parameter values are identical across cell lines, we applied the trained model parameters constrained by time series data of pAkt, pERK, and pc-Fos in the MCF-7, BT-474 and MDA-MB-231 cell lines to the SK-BR-3 cell line model (Figure 4a). By using its autologous gene expression data and the trained parameters of the other three cell lines, we succeeded in predicting experimental observations for SK-BR-3 (Figure 4b). This result suggests that expression levels of the genes in the ErbB network quantitatively control their temporal signaling dynamics, and that this principle is universal across breast cancer subtypes.

### 2.5. Sensitivity Analysis of Initial Values for the SK-BR-3 Cell Line

To examine how a change in the amount of an ErbB network component (e.g., gene expression level) affects cellular output, we performed sensitivity analysis [38] for initial values (amount of proteins or molecules) in the SK-BR-3 cell line. Because phosphorylated Akt and c-Fos are the final outputs of the signaling pathways implemented in the model and play an important role in cellular decision making [39], we focused on them and quantified their cumulative responses as the integral over the observation time of 120 min. The sensitivities were calculated by varying the amount of each nonzero species and simulating pAkt and pc-Fos responses. 

We visualized sensitivities for each parameter set, not the averaged sensitivities. Akt activity, which is often associated with drug resistances in breast cancer cells [40], is strongly controlled by the amount of PI3K and PIP2 (Figure 5a,b) in both EGF and HRG conditions. We found that adaptor proteins, such as Shc, Grb2 and Gab1, generate both positive and negative effects on the pAkt response, depending on their abundance, with good reproducibility (Figure 5a,b). This result may imply that the stoichiometry of the adaptor proteins is critical in proper regulation of Akt activity. The sensitivities on cumulative pc-Fos responses showed that kinases of the MAPK pathway (Raf, MEK and ERK) are critical for the phosphorylation of c-Fos (Figure 5c,d). Though experimental data showed similar time course patterns of pc-Fos in stimulation by both growth factors, the control coefficients implied that Grb2 and SOS are the potential positive regulators for the pc-Fos response only for HRG (Figure 5d).

## 3. Discussion

In this study, we presented a computational platform for model building and numerical analysis of cancer signaling networks. We used RNA-seq data to infer the initial values of model components and numerous time course phosphorylation data (pAkt, pERK, and pc-Fos) for training the model parameters. Using this platform, we constructed a comprehensive model of ErbB receptor signaling-early transcriptional regulation by integrating two models independently developed in previous studies [14,20]. Several mathematical models for ErbB signaling pathways have been already published [9,10]. However, our model is valuable as it includes early transcriptional regulation by ErbB1-4 receptors. The early transcriptional products such as c-Fos and c-Myc [13] connect ErbB signaling to a variety of other biological process, including cell cycle [16,17] and metabolism [19], which play critical roles in cancer physiology. We expanded the earlier c-Fos regulation model [14] to include the interactions between growth factors and membrane ErbB receptors through the intracellular signaling cascade [20]. The obtained model allows us to investigate the quantitative mechanisms of ErbB receptor control over early gene regulation. Using the BioMASS framework, the model can be flexibly extended to add more experimental data and biological information, re-tune the parameters and identify the network mechanisms at any resolution. Our computational framework can be used for inferring signaling dynamics and underlying mechanisms in a wide variety of signaling networks by using proteomics data [41] or small-scale Western blot data, together with RNA-seq data. 

The methods developed by Hidalgo et al. [42] and Cubuk et al. [43] successfully captured phenotype-specific pathway alternations in signaling and metabolism, respectively. However, these models simplified all post-translational modifications as either activation or inhibition. The framework proposed in this study allows us to handle more detailed mechanistic models and therefore to analyze biologically relevant targets for manipulations of the system. As showing in the sensitivity analysis, large scale mechanistic modeling helps to extract important insights that cannot be achieved by statistical modeling or machine learning methods alone [44].

Using the CCLE RNA-seq data as the initial values for each cell line, the model, which is jointly parameterized by training data sets of several cell lines, gained an additional predictive ability to infer signaling dynamics in a different cell line. This result indicates that gene expression levels of ErbB network components play an important role in the control of signal transduction dynamics. It also indicates that gene expression data rather than mutation data, which often indicate changes in enzyme activity of kinases and phosphatase or protein binding affinity, may be suitable for inferring signaling dynamics and activities. Previous studies have shown that signaling dynamics control cell fate [45,46], but it has remained unclear how different genetic information is encoded in different dynamic patterns of signaling molecules. Our study suggests an important hypothesis, that the gene expression level is critical to shape the signaling dynamics and forms the basis of regulatory mechanisms of cancer. Further work will be needed for validation of this hypothesis and to determine whether this principle is conserved in other cell types from different tissues.

In this report, we showed an example of sensitivity analysis for a particular cell line SK-BR-3. The sensitive process, which is usually a bottleneck in the network, is cell-specific. Therefore, comparison of the sensitive regions among different cell lines will be necessary to generate additional information about network homeostasis. It has been reported that a particular gene expression profile could improve the classification of breast cancer and prediction of therapeutic responses [47]. We anticipate that our data-driven modeling platform based on clinical genomic and transcriptomic data [48] will be a useful tool for the construction of personalized patient-specific models [49]. An in silico patient model will have predictive power to generate clinically valuable information by integrating and analyzing heterogeneous clinical data in a quantitative manner, therefore it should help to optimize disease treatment strategies and improve the prognosis.

## 4. Materials and Methods 

### 4.1. Model Simulation and Parameter Estimation

We constructed the mathematical model linking ErbB receptor to c-Fos regulation by integrating two previously developed mathematical models. The Birtwistle model [20] and the Nakakuki model [14] accounted for the processes from the activation of ErbB receptors to ERK phosphorylation and from MEK phosphorylation to c-Fos induction, respectively. These models contain overlapping regulation in the MAPK cascade and are integrated so that phosphorylated MEK triggered by ErbB receptors can activate downstream ERK for c-Fos transcription. The mathematical formula describing the integrated model can be found in Method S1. We described the biochemical reactions using ODEs. To train model parameters, i.e., kinetic constants and weighting factors to obtain initial values against time series phosphorylation data from three breast cancer cell lines (MCF-7, BT-474, and MDA-MB-231), we minimized the sum of squared differences between the experimental observations and the simulated values using the genetic algorithm adapted from DIDC [33]. We obtained the parameters from the original models and re-searched the parameters after model integration. The searched area was 0.1 to 10 times the original values. The search range of the weighting factor, which is explained below, was from 0.1 to 100. Information about the objective function to be minimized and search parameters can be found in fitness.py (.jl) and set_search_param.py (.jl), respectively (Table 1). Differential equations were integrated using Sundials.jl [50,51] for training and scipy.integrate.ode [52] for validation and sensitivity analysis.

### 4.2. CCLE Data

CCLE RNA-seq gene expression data used for the model individualization were downloaded from https://data.broadinstitute.org/ccle/CCLE_RNAseq_rsem_genes_tpm_20180929.txt.gz. Initial protein levels of the model species (total of 19 genes: ErbB1, ErbB2, ErbB3, ErbB4, Grb2, Shc, PI3K, RasGAP, SOS, Gab1, Akt, RasGDP, Raf, MEK, PTP1B, CREB, ERK, Elk1, and RSK) were inferred from gene expression data (See the detailed gene list in Appendix A). Previous studies have shown that the correlation between protein and transcript levels is relatively low [53,54] and that there is a gene-specific translation mechanism. We assumed that genes belonging to the same family have a redundant protein function. Therefore, we inferred the protein level from one or more genes in the same gene family (isoforms) and adopted the weighting factor to convert the TPM value to the appropriate initial protein value in the model. The corresponding weights for each gene (weighting factors), were assumed to be conserved across cell lines and were included in the search parameters. To extract gene expression datasets from CCLE, we developed a simple tool, ccle_extractor (https://github.com/okadalabipr/ccle_extractor), a Python script for processing CCLE datasets.

### 4.3. Cell Culture and Western Bloting

MCF-7, BT-474, SK-BR-3 and MDA-MB-231 cells were maintained in Dulbecco’s modified Eagle’s medium (DMEM) supplemented with 10% fetal bovine serum (FBS). Before treatment with 10 nM EGF or HRG, the cells were synchronized by serum starvation for 16 h. Cells were lysed with BioPlex Lysis buffer, cell lysates were cleared by centrifugation and total protein concentration in supernatants was determined using a protein assay reagent from Bio-Rad Laboratories, CA, USA. For Western blotting, anti-phospho-Akt (Thr308, no. 2965) and anti-phospho-ERK (Thr202/Tyr204, no. 4370) were purchased from Cell Signaling Technology (Santa Cruz, CA, USA). Anti-phospho-Fos (Ser374, ab55836) was purchased from Abcam (Cambridge, MA, USA). We adopted the transfer and normalization methods described in previous studies to minimize transfer errors and variability between the blots [55,56]. Protein band intensities were quantified using Fiji [57]. The data were normalized between minimum (0) and maximum (1) values. All raw data is available in Appendix A.

### 4.4. Data and Code Availability

All data and code for model simulation, parameter estimation and prediction are available from the GitHub website: https://github.com/okadalabipr/Imoto_Cancers_2020. BioMASS is available from https://github.com/okadalabipr/biomass.

## 5. Conclusions

We constructed a comprehensive model of the immediate-early gene response triggered by ErbB receptor activation and predicted signaling and transcriptional activities based on RNA-seq data of breast cancer cell lines. We developed a computational framework for model construction, parameterization and sensitivity analysis of the model. Our study indicates that gene expression levels in the ErbB network may govern the cell-specific signaling dynamics across breast cancer cell lines and therefore play an important role in regulation of cell fate.

## Figures and Tables

**Figure 1 cancers-12-02878-f001:**
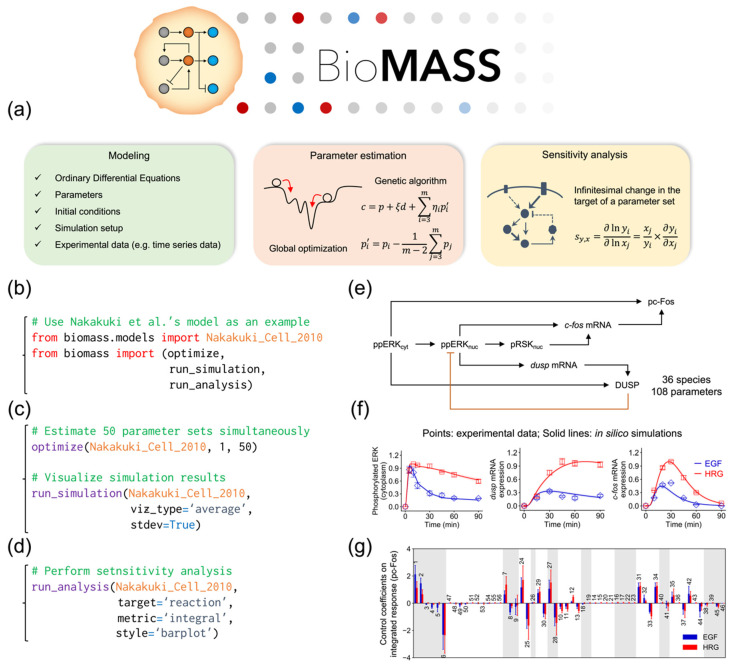
Overview of the BioMASS workflow. (**a**) After setting model properties (see Table 1), users can estimate unknown model parameters against experimental observations. Based on optimized parameter sets, users can perform sensitivity analysis to identify critical components or processes for cellular output. This workflow implements the model of Nakakuki et al. (**b**–**d**) Example code for model import (**b**), parameter estimation and visualization of simulation results (**c**) and sensitivity analysis (**d**). (**e**) Core reaction scheme of the Nakakuki model. (**f**) Simulation results using optimized parameter sets. (**g**) Output of the sensitivity analysis.

**Figure 2 cancers-12-02878-f002:**
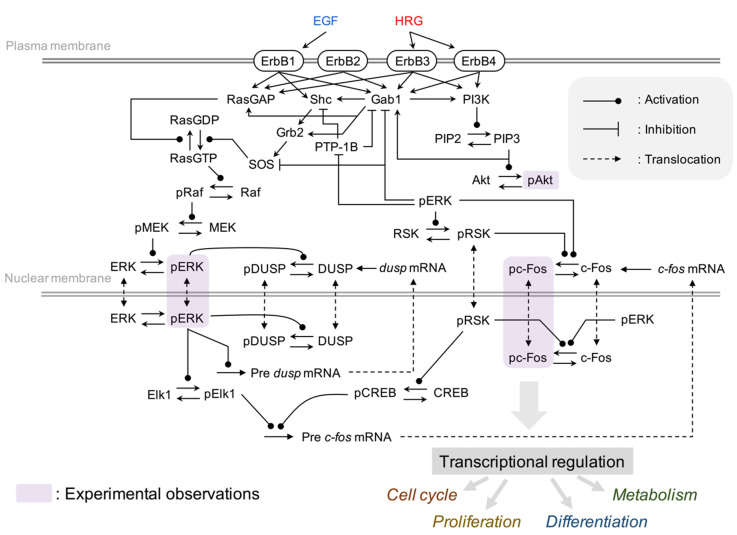
Influence diagram of the molecular interactions implemented in the model.

**Figure 3 cancers-12-02878-f003:**
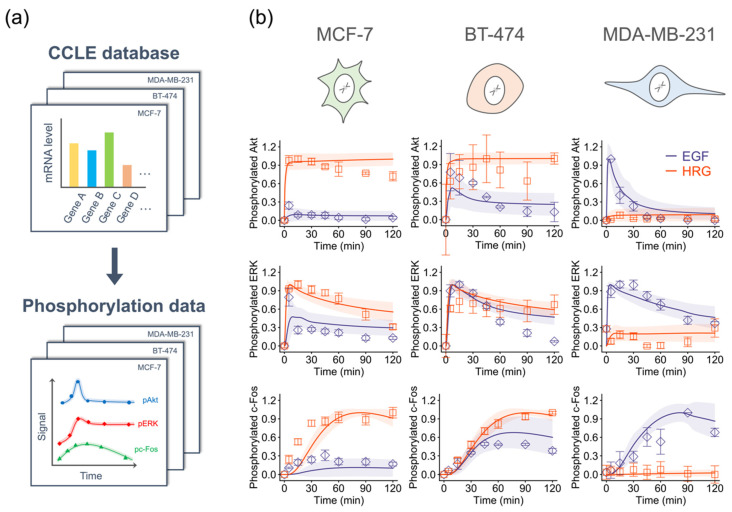
Parameter training against experimental datasets. (**a**) Overview of the parameterization procedure. (**b**) Results of parameter estimation. Points: (Blue diamonds, EGF; red squares, HRG) denote experimental data and lines denote the averaged simulation results of 30 parameters, and shaded areas indicate standard deviation. For all panels, error bars denote standard error for three independent experiments. The Western blot images can be found in Appendix A.

**Figure 4 cancers-12-02878-f004:**
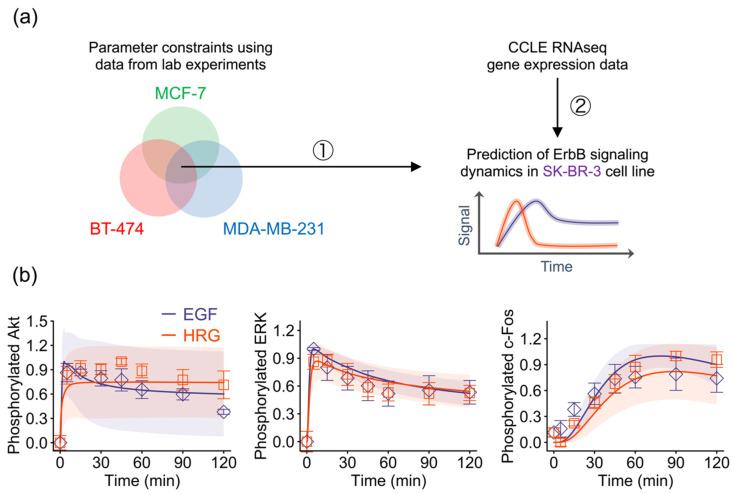
Prediction of ErbB signaling dynamics in the SK-BR-3 cell line. (**a**) Strategy for model-based prediction. (**b**) Predicted signaling dynamics and experimental observations. Points: Experimental data (blue, EGF; red, HRG); solid lines: Averaged in silico simulation results of the 30 parameters; shaded areas: Simulation standard deviation.

**Figure 5 cancers-12-02878-f005:**
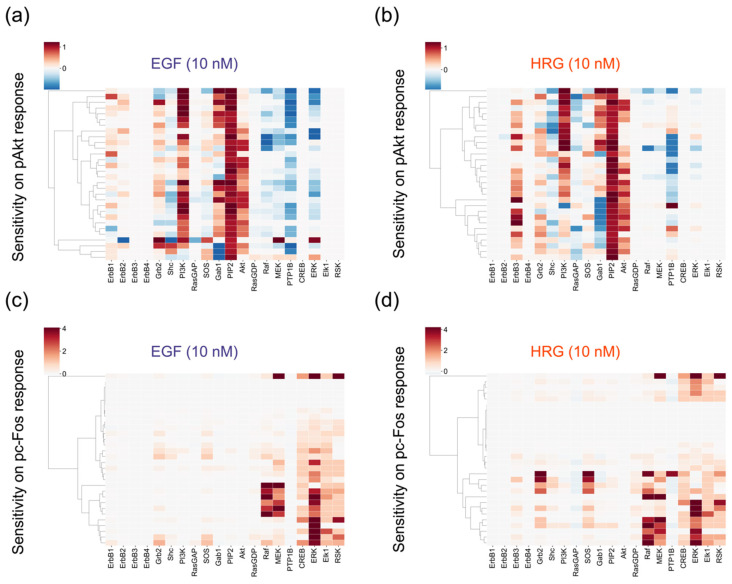
Results of sensitivity analysis. Negative coefficients (blue) indicate that the quantity of the response metric will decrease when the amount of a species increases, while positive coefficients (red) indicate that the metric will increase. (**a**) Epidermal growth factor (EGF)-induced pAkt response. (**b**) heregulin (HRG)-induced pAkt response. (**c**) EGF-induced pc-Fos response. (**d**) HRG-induced pc-Fos response.

**Table 1 cancers-12-02878-t001:** A brief description of each model file/folder in BioMASS.

Name	Content
name2idx/	Names of model parameters and species
set_model.py	Differential equations, parameters and initial conditions
observable.py	Observables, simulations and experimental data
viz.py	Plotting options for customizing figure properties
set_serach_param.py	Model parameters to optimize and search region
fitness.py	An objective function to be minimized
reaction_network.py	Reaction indices grouped according to biological processes

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
