# Peer review of "A Computational Framework for Prediction and Analysis of Cancer Signaling Dynamics from RNA Sequencing Data—Application to the ErbB Receptor Signaling Pathway"

_cancers, 2020, doi:10.3390/cancers12102878_

Round 1

Reviewer 1 Report

In this work, the authors present a new computational method for predicting signaling dynamics based on gene expression data. This method involves the creation of a differential equation model of a relatively large-scale signaling network, determining the parameters of that model by training it on data from known breast cancer cell lines, then using the trained model to predict behavior of a different cell line (on which the model had not been trained). The results show that the optimization from the training cells resulted in a model that could predict behavior of a cell line to which the model had not been exposed.

A key finding is that the model parameters obtained by training on other cell types appear to work well when the model is asked to predict behavior in a different cell. This indicates that the parameters of interactions of molecules in these different cell types are not particularly unique to the cell type and strongly implies that the expression levels of the various members of a signaling network is sufficient for describing the mechanism of action in complex signaling networks.

While I’m not sure that this really represents “a new computational framework,” I do find this result to be interesting and useful as, in my opinion, it makes a solid contribution to the discussion as to the need for quantitative (as opposed to qualitative) modeling of complex signaling networks.

I believe that it is entirely appropriate that studies of intracellular signal transduction continue to consider networks of increasing size and complexity as there is considerable evidence that there are functions of these networks that are lost when a reductionist approach is taken. However, modeling large-scale systems can be a challenge, particularly with continuous methods (e.g., ODE’s) as the problem of parameter estimation increases with the size of the network and solutions of the equations over longer time scales can become intractable.

Discrete methods solve the intractability problem but are open to criticism because they do not take into account the parameters of the actual interactions of various components of a network. While the parameters of interaction are generally considered to be critical in modeling the interactions of a low number of components, it’s not clear that they are as important when looking at a large number of simple interactions (which describes many signaling networks that work through phosphorylation-activated enzyme cascades).

In this paper, the authors show that for the relatively large-scale network under consideration, the parameters of the interactions recede in importance as far as modeling/predicting behavior. Instead, the dynamics of this network appear to be mostly dependent on the levels of expression of the interacting components. Thus, these results predict that a parameter-free, discrete model could have been built that would be able to reproduce the phenomena of this signaling network—an important finding.

As mentioned by the authors in the discussion, there are a number of avenues of inquiry that this paper might suggest. E.g., building a discrete model (relatively easy to do these days as there are a fair number of tools available for doing this) and confirming that it could produce the same results.

As I mentioned above, I’m not sure I agree that this represents “a novel computational method” for predicting signal transduction, but neither am I certain of this as I suppose it is open to interpretation. In my mind, there are already parameter-independent ways to model and predict signaling network behavior, but the question is whether they are valid since they don’t take parameters into account. I think this paper says something important about that question rather than creating a new method. However, this is not a large enough concern to require a revision.

A significant positive is that I found this paper to be refreshingly short, focused, and easy to read. I also found no typos or grammar problems. As such I do not suggest any major revisions.

The only minor revision I would submit for consideration is in regard to figure 4—I’m not seeing the “shaded areas” in the figures that are mentioned in the legend. I was also curious as to whether loading controls were used in the western blots for normalizing beyond the 0 to 1 normalization that is mentioned.

In summary, this is a highly-focused paper that does not suffer from over ambition, and it presents a nice finding that says something useful about the question of the need for interaction parameters when modeling large-scale signaling networks.

I recommend it for publication with no need for any major changes.

Author Response

We would like to thank the Reviewer for her/his positive comments. According to the suggestion, we simply call it “a computational framework”, not “a new computational framework”.

In the case of Figure 4, some computers do not seem to see the shaded area. The figure has been updated.

For normalization of western blot data, the experimental values range from 0 to 1 because the band intensities are normalized using the maximum and minimum values in time-course. The molecular weights of the ERK and cFos proteins are closely overlapping with loading controls such as GAKDH and actin, therefore they are not used.

Reviewer 2 Report

The authors describe a new model that can estimate the signaling dynamics (phosphorylation rates) using gene expression values.
The study is timely important since it is touching the field of mechanistic pathway analysis that is the emerging field of systems biology.
The manuscript has written clearly but while reading I find diversification in the objective of the study.
The objective of the model proposed needs to be clearly stated and which gaps are being filed in the systems biology with the usage of BioMASS.
Since I am not able to open the supplementary files, some of my comments may be addressed in the supplementary material.

Below you can find my comments;

1) While the title is very generalized, the manuscript itself focuses only ErbB - cfos pathway.
It seems to me that the model can only be applied to this pathway. It will be more understandable if the manuscript contains description of a
generalized method and additionally a case application as proof of concept (ERBB pathway results)
Otherwise, the title needs to be rearranged to state that this is a specific model for only ERBB pathway.

2) It would be great to see different examples of BioMASS application which can show that the proposed model is not only compatible with signaling pathways that are composed of only phoshorilation.

Pathways are composed of several modifications (ubiquitination, glycosylation, etc.).
Does the model still able to predict phosphorylation patterns correctly from gene expression when a pathway is regulated by different post-translational
modifications?

Does the model use the entire pathway topology? Is there any propagation method applied? How the feedback loops are converged?
It is hard to do imagine what is behind the model proposed without seeing mathematical formulas.

3) How the "ErbB receptor → c-Fos induction" pathway was constructed? Is it an in-house and new network or taken from a pathway database?

4) The authors trained a model with kinetic parameters and RNAseq data. Are the kinetic parameter obtained from phosphorylation data?
As I understand, RNAseq was taken from CCLE. How the kinetic parameters were obtained, also from CCLE?
Is there a machine learning algorithm used for the training purpose? Could you please explain in more detail the training method and the datasets used?

Also in Line 79: "Then, the model proceeded to prediction in another cell line on which it
was not trained, using its own phosphorylation data. The model learned the parameters from other cells lines and was able to accurately predict the quantitative behaviour of the untrained cell line solely from its autologous gene expression data."

For the prediction purpose do you use phosphorylation or RNA-seq data?

5) Did the authors compared the predictive power of gene expression versus the rate constants of kinase or phosphatase or protein-protein interactions to conclude such
strong conclusion as it given in line 25: "Overall, our study suggested that gene expression levels in the ErbB network, not the rate constants of any kinase or phosphatase, or protein-protein interactions, can explain the cell-specific signalling dynamics in breast cancer and therefore play an important role in determining cell fate."

6) Line 50: "Upon ligand binding, the receptors form dimers, activate their intrinsic tyrosine kinases and stimulate multiple signaling pathways".

With the multiple signalling pathways, does the author refer to different mechanisms or sub-pathways in ErbB pathway or refers to crosstalk between pathways.
Or does these receptors also regulate different pathways?

7) Line 53: Does the following information specific to the cell line mentioned? While the information before and after this sentence looks quite general, in this line suddenly it becomes cell line-specific.

"The MCF-7 breast cancer cell line, EGF and heregulin (HRG), another member of the EGF family, induce transient and sustained ERK activity, respectively."

8) Line 55: "Ultimately, the difference in ERK activation dynamics changes expression patterns of immediate-early genes, such as c-Fos and c-Myc transcription factors, which play a central role in controlling cell cycle, differentiation, and metabolism (***cellular functions)."

Shall we see the c-Myc in the Figure 2? I am not able to localize it.
I recommend adding these ***cellular functions to the pathway layout in Figure 2.

e.g: Effector Gene A --> Cellular function X.
You can also check https://www.genome.jp/kegg-bin/show_pathway?hsa04012.
The effector proteins, the proteins at the very end of each cascade are linked with the cellular functions.

Line 178: "Because phosphorylated Akt and c-Fos are the final outputs of the signaling pathways implemented in the model.."

Maybe you can highlight these genes in Figure 2. It will increase the interpretation of the modeled pathway. Otherwise, this figure looks like a loop of several feedback loops.

9) Line 57: "Therefore, a system-level understanding of the common mechanisms that regulate ErbB dynamics across cell types is important for uncovering a fundamental mechanism in cancer progression."

Line 60: "In this study, we constructed a comprehensive mathematical model of the ErbB receptor to c-Fos activation..."

For me it is unclear if this manuscript focuses on the analysis of ErbB pathway or uses analysis of  this pathway as the proof of concept.

10) Line 149: "... as initial values of the 20 genes described in the
model (Figure 3a)."

Figure 3a is a schematic description of this sentence.
I believe that it is more interesting to see the list of 20 genes.
Therefore "Figure 5" can be added in the text, near "20 genes" inside the parenthesis.
You can also mention that these 20 genes are the same genes shown in Figure 2 (if this i the case).

In the legend of Figure 5, I am not able to understand the following part; "quantity will decrease when the amount of a species increases".

Does quantity mean gene expression? What is the species?
Why there is PTP1B gene in Figure 5 but not in Figure 2?

11) Line 167: "Since we assumed that parameter values are identical across cell lines ..."
Can the author support this assumption with some scientific references?

13) Line 180: "...we focused on them and quantified their cumulative responses"
Line 182: "The sensitivities were calculated by varying the amount of each
nonzero species.." 

Does line 180 refer to the western assays? What are the nonzero species?

14) Which tables or figures are supporting the following results?

"Line 185: We found that adaptor proteins, such as Shc, Grb2 and Gab1, generate both positive and negative effects on the pAkt response, depending on their abundance, with good reproducibility."

15) Line 246: "Initial conditions of the model species (total of 20 genes) were determined from gene expression data, we multiplied TPM values by weighting factors."

I am not sure if it is right way to call cell lines as model species. What are the weighting factors? How they were obtained and why this multiplication was needed?

16) The authors mentioned the limitations of the comprehensive models and why mathematical modeling has emerged as a powerful method.
The mechanistic pathway analysis is one of the novel improvements in the field of systems biology. Even the most recent methods only accept gene expression and/or genomic data to analyze pathways. It is worth to mention these recent improvements (examples are below) and how your method fills the gaps remained from them. It will help the readers  understand what is new in your study and how BioMASS will boost the improvements in the systems biology. The methos developed by Hidalgo et.al and Cubuk et.al (https://pubmed.ncbi.nlm.nih.gov/28042959/ and https://pubmed.ncbi.nlm.nih.gov/30135189/) use gene expression data. And using simple models (gets the power from gene expression and the pathway topology) they are able to capture phenotype spefic pathway alterations in different biologocal concepts; signaling and metabolism.
However the limitations of these models are simplification of all the post translational modifications into activation and inhibition. BioMASS offers more detailed analysis. It removes the limited proteomics data bariers and make the more realistic
pathway activity simulations possible. The predictions of BioMASS can help development of new mechanistic pathway activity approaches. The new approaches that use postranslational modification measures (estimates) obtained through BioMASS....

17) Line 18: "In this study, we introduce a novel computational method to predict the signaling dynamics from RNA sequencing (RNA-seq) gene expression data."

Line 208: "Our computational framework can be used for inferring signaling dynamics and underlying mechanisms in a wide variety of signaling networks by using proteomics data [39] or small-scale western blot data, together with RNA-seq data."

For the method proposed what are the required input types?
Only RNA-seq or RNA-seq plus some other omics.

18) Could you please give more details in the section "Model Simulation and Parameter Estimation"?

As you clearly mention how CCLE data extracted. You can also indicate which python code (from Table 1) used for each step.

19) Line 269: "Our study indicates that gene expression levels in the ErbB network govern the cell-specific signaling dynamics across breast cancer cell lines and therefore play an important role in determining subsequent cell fate."

How the authors able to get the conslusion about the cell fate? From which results we can get the same conclusion?

Author Response

First of all, we thank the Reviewer for her/his constructive and helpful comments. We think that the manuscript has been greatly improved now. All responses are included in the attached file.
